# The Stimulation of Neurogenesis Improves the Cognitive Status of Aging Rats Subjected to Gestational and Perinatal Deficiency of B9–12 Vitamins

**DOI:** 10.3390/ijms21218008

**Published:** 2020-10-28

**Authors:** Grégory Pourié, Nicolas Martin, Jean-Luc Daval, Jean-Marc Alberto, Rémy Umoret, Jean-Louis Guéant, Carine Bossenmeyer-Pourié

**Affiliations:** Faculté de Médecine, INSERM 1256/University of Lorraine, F-54500 Vandoeuvre-les-Nancy, France; nicolas.martin@univ-lorraine.fr (N.M.); jean-luc.daval@univ-lorraine.fr (J.-L.D.); jean-marc.alberto@univ-lorraine.fr (J.-M.A.); remy.umoret@univ-lorraine.fr (R.U.); jean-louis.gueant@univ-lorraine.fr (J.-L.G.); carine.pourie@univ-lorraine.fr (C.B.-P.)

**Keywords:** neonatal B-vitamins deficiency, neurogenesis, hippocampus, behavior, learning, long term follow-up

## Abstract

A deficiency in B-vitamins is known to lead to persistent developmental defects in various organs during early life. The nervous system is particularly affected with functional retardation in infants and young adults. In addition, even if in some cases no damage appears evident in the beginning of life, correlations have been shown between B-vitamin metabolism and neurodegenerative diseases. However, despite the usual treatment based on B-vitamin injections, the neurological outcomes remain poorly rescued in the majority of cases, compared with physiological functions. In this study, we explored whether a neonatal stimulation of neurogenesis could compensate atrophy of specific brain areas such as the hippocampus, in the case of B-vitamin deficiency. Using a physiological mild transient hypoxia within the first 24 h after birth, rat-pups, submitted or not to neonatal B-vitamin deficiency, were followed until 330-days-of-age for their cognitive capacities and their hippocampus status. Our results showed a gender effect since females were more affected than males by the deficiency, showing a persistent low body weight and poor cognitive performance to exit a maze. Nevertheless, the neonatal stimulation of neurogenesis with hypoxia rescued the maze performance during adulthood without modifying physiological markers, such as body weight and circulating homocysteine. Our findings were reinforced by an increase of several markers at 330-days-of-age in hypoxic animals, such as Ammon’s Horn 1hippocampus (CA1) thickness and the expression of key actors of synaptic dynamic, such as the NMDA-receptor-1 (NMDAR1) and the post-synaptic-density-95 (PSD-95). We have not focused our conclusion on the neonatal hypoxia as a putative treatment, but we have discussed that, in the case of neurologic retardation associated with a reduced B-vitamin status, stimulation of the latent neurogenesis in infants could ameliorate their quality of life during their lifespan.

## 1. Introduction

Methyl donor deficiency during the perinatal period is known to induce dramatic structural and functional consequences in various organs of newborns. Some of the most studied in neonatology are the spina bifida syndrome especially affecting the nervous system development and function. Different stages of gravity are known and even if some of them allow survival and growth of infants, numerous nervous functions remain impaired [1,2,3]. These dramatic outcomes lead to supplementation of pregnant women with folate (vitamin B9) in numerous countries, or diet fortification in others [4,5]. Moreover, vitamin B12 injections represent the only proposed treatment in cases of specific diseases, such as rare genetic mutations in B12 metabolism [6,7,8]. Nevertheless, except for the efficient correction of the methyl donors (B-vitamins) deficiency during pregnancy, numerous cases of B9/B12 metabolism deregulation during the first years of life still lead to long lasting neurologic and behavioral dysfunction [9,10,11].

Actually, a mechanism described as “fetal programming” or “Barker theory” predicts that if a disruption occurs during the development and maturation of a cellular circuit or an organ, the functional outcomes will be dramatically disturbed during the lifespan [12,13,14,15]. This was shown for the brain in case of micronutrients and methyl donor deficiencies [16,17,18,19]. Nevertheless, the brain has been shown to be plastic for either its structure (i.e., number of neurons and/or synapses) or its function (i.e., variety of synapse functioning or behavioral adaptation) [20,21]. These forms of plasticity could be activated though different mechanisms. Among them, neurogenesis represents a latent biologic response that can be up-regulated by environmental factors for instance [22]. We and others have shown that, surprisingly, a short hypoxia occurring during the early natal period is able to activate neurogenesis in specific permissive brain areas and leads to rescue of the structural and functional outcomes of important neuronal circuits such as the cognitive areas [23,24,25,26,27]. Thus, we proposed to use a short and transient neonatal hypoxia as a stimulus in order to investigate whether a brain damaged by a B-vitamin deficiency in pups could respond in a positive way in terms of neurogenesis and/or plasticity. A linked objective was to follow the cognitive capacities of growing pups until adulthood and throughout aging, and to investigate tissue and molecular markers in relation with the putative plastic reaction of the brain. Our global objective was not to propose a treatment with hypoxia in the case of positive results, but to highlight putative brain reactions and propose molecular targets in order to complement the classical B-vitamin treatments.

## 2. Results

Previous studies have already shown that a deficiency in B-vitamins during gestation and lactation leads to reduced one-carbon metabolism markers and global body status in the young [16,28]. It has been documented also that a restoration of the metabolic status could partly compensate the growth retardation [16]. Nevertheless, our results showed that after a switch to a normal diet at the weaning period, male and female pups did not follow the same pattern of growth. As reported in Figure 1, actually, at D40, males subjected to either the deficient diet (MDD males) or the deficient diet with the neonatal hypoxia (MDD-H males) showed the same body weight as control males. This compensation appeared to be maintained until adult life in males (Figure 1A). However, females did not follow such a restoration of body weight during adulthood since significantly lower body weights were measured for MDD and MDD-H females compared to controls at each age-point from D40 to D205. Nevertheless, at D330, only MDD-H females presented a lower body weight than controls (not MDD females, Figure 1B). Measuring the biochemical status in plasma at D80 and D330, we observed a normalization of each marker in both males and females with the folate between 250 ± 62 to 170 ± 59 nmol/L and vitamin B12 between 812 ± 182 to 453 ± 96 pmol/L. We measured an age-related decease of these marker from D80 to D330 in all animal categories but not within groups.

Testing learning and memory capacities of young rats using a multiple-T maze, we also obtained a gender effect. Males were less affected than females by the deficient diet and hypoxia, since a higher escape latency was measured for MDD males at S1 during the five learning sessions (Figure 2A). In females, the deficient diet induced an elevation of the escape latency in all sessions of learning (S1 to S5, Figure 2B), whereas the neonatal hypoxia compensate such an effect, except for session one (S2 to S5, Figure 2B). When the memory retrieval was tested long after the end of the learning period before 50-days-of-age, one more time a gender effect appeared. Actually, males showed an increase of the escape latency whatever the groups from D80 to D330, and neither the initial diet nor the neonatal hypoxia appeared to modulate this result, except for D80 (Figure 2A). In females, we showed that the initial deficient diet led to an increase of the escape latency related to memory retrieval, whereas the neonatal hypoxia was correlated to comparable results as controls at D80 and D330 (no significant results were measured at D205, Figure 2B). As a comparison between males and females for memory retrieval, it appeared that males presented a time-related decreased memory capacity while the exit parameter increased, when females rather showed a stable memory until older-ages (only the escape latency in the maze was presented, but other parameters, not presented, such as the number of errors committed also attested the same results).

Exploring the metabolic status at D330, we obtained an elevation of plasma homocysteine levels in all females subjected to the deficient diet (with the hypoxia treatment or not) compared to controls. Old males did not present such a difference (Figure 3A). In order to find correlations between the poorer memory retrieval of MDD females and the hippocampus status, we measured CA1 neuronal layer thickness at D330, long after the initial deficient diet and neonatal hypoxia. It appeared that only MDD females presented a significant reduction of this neuronal layer, since all males groups and MDD-H females did not (Figure 3B).

## 3. Discussion

Several previous studies have shown that brain development is affected by a gestational and neonatal deficiency in B-vitamins, also named methyl-donor deficiency. Thus, the tissue status and/or the functioning of various brain areas could be dramatically reduced or modified [16,23]. In humans, the correction of circulating levels of B-vitamins (folate and B12) leads to non-satisfying restoration of neurologic scores despite biochemical and physiological compensations [6,7,9]. In addition, a deceased status in methyl donors is correlated to accelerated cognitive decline and senescence [29,30,31]. Actually, studies using animal models showed the efficiency of supplementations on plasma and body parameters, attesting to a restoration of healthy conditions for animals initially subjected to various methyl-donor deficiencies [28,32,33,34]. Nevertheless, at least two questions remain unclear. Why classical corrections of the metabolic parameters (i.e., folate and/or B12 supplementations) are poorly efficient for brain damage in adults [35], and secondly, is it possible to find another route to avoid or rescue neurologic disorders? And finally, as related outcomes, what are the long-term consequences of the affected neuronal capacities?

### 3.1. The Correction of B-Vitamin Deficiency Did Not Lead to the Rescue of All Body Parameters

From a metabolic point of view, the correction of vitamin status and/or the restoration of a non-deficient diet could efficiently restore the circulating physiological parameters such as folate and vitamin B12 concentrations, as well as for the related markers, S-adenosine methionine, S-adenosine homocysteine, homocysteine, and methyl malonic acid. These adjustments are correlated in the majority of cases to a body weight restoration in initially deficient animals compared to controls [16]. In our model, rat pups that were submitted to a B9-B12-deficiency during the gestation and lactation, received a switch to a normal non-deficient diet at weaning time (post-natal day 21). As a physiological consequence, our results showed also such a healthy restoration of body parameters during adulthood after the diet correction, but we obtained a gender effect. Actually, deficient females presented a persistently lower body weight until older ages, as it was the case during the deficient period. This result correlates with another study exploring modulations of folic acid status [36]. Using neonatal hypoxia as a method to promote latent neurogenesis, we first observed that hypoxia did not modify this global body status neither in males nor in females. Thus, females appeared more sensitive than males through a long period of data analysis. The implication of epigenetic modifications and/or physiological mechanisms such as estrogen receptor transduction could explain such a difference between males and females [37,38].

### 3.2. The Initial B-Vitamin Deficiency Provokes Persistent Cognitive Deficits Corrected by the Interventional Stimulation of Neurogenesis

Since it is documented that neurological defects, and especially psychiatric and cognitive problems, correlate with a deficient status in B-vitamins [39,40,41], we proceeded to a long-term monitoring of hippocampus-dependent learning and memory functions. The learning capacities of rats were tested with a navigation test in a multiple T-maze, and as a comparison with body weight evolution, males were less affected than females by the neonatal deficiency. The learning test was conducted with five successive sessions between days 14 and 18 within the deficient period. One more time a gender effect appeared, since deficient females presented lower performances at all sessions compared to their controls, whereas deficient males did not, except for session one. However, it appeared that these functional deficits in females were not a dramatic and constant neurological disorder. Actually, deficient females presented a decrease in the escape parameter, attesting to learning acquisition; so we rather conclude to a neurologic retardation in growing females instead of inoperative neuronal functioning. The methodological use of a mild transient neonatal hypoxia reversed these lower learning performances in deficient females. As it was already shown, the interventional stimulation of latent neurogenesis with a mild transient hypoxia explained such a rescue [23,24]. Evaluating the memory retrieval long after the end of the learning procedure, we obtained an expected “syndrome of forgetfulness” since both males and females presented higher escape parameters compared to the last learning session. Nevertheless, deficient males presented poorer retrieval performances in the same multiple T-maze than controls and hypoxic males at day-80. However, this characteristic disappeared with males getting older, suggesting a senescent evolution of all male groups, even non-deficient ones as it has already been shown in rodents or humans [42]. Considering adult females between days-80 and 330, the initial deficient status was correlated with constant poorer memory performances (except at day-205), while the hypoxia status led to comparable results with controls. This shows that (i) females did not present a senescent evolution during the examined period compared to males (i.e., until 330-days-of-age), and (ii) the beneficial effects of hypoxia-related mechanisms in the brain appeared persistent during adulthood in females. Such a gender susceptibility has been shown especially for neurobehavioral functions during aging in rats [43].

### 3.3. Various Tissue Parameters Attested of a Rescue Mechanism

In order to explain these functional outcomes, we explored various parameters at 330-days-of-age, starting with the plasma homocysteine, since it was already presented as an associated factor with neurodegenerative diseases, especially affected specific brain areas such as the hippocampus [16,44]. We found that, surprisingly, homocysteine levels remained higher in older females initially subjected to a deficient diet (with not effect of hypoxia or male status). Despite this plasma levels appeared moderately elevated rather than pathogenic (i.e., between 5 and 7 µmol/L rather than over 10 µmol/L), it could nevertheless attest to a persistent modified metabolic status in aging females compared to males. However, since females subjected to short hypoxia presented both such an elevated homocysteine level and at the same time a restoration of cognitive performance, these parameters could not be directly related. Thus, we explored other parameters such as those related to brain circuit plasticity at 330-days-of-age. The CA1 hippocampus layer, usually presented as a tissue marker for hippocampus health, appeared reduced in thickness in deficient females expressing poorer cognitive performance. At the same time, hypoxia appeared to conserve comparable CA1 thickness compared with controls indicating that a mechanism for circuit protection or stability occurred, despite the early deficient event. Such a mechanism was already shown in the same deficient and hypoxic conditions but in younger ages [23]. The conditions of the short hypoxia in young individuals have been described already and lead to enhanced brain protection especially through neurogenesis; this plastic biological response could be named “hypoxia-stimulated-mechanism” [24,25]. Thus, it seems that the mechanism stimulated by a short hypoxia could compensate the already documented deleterious effects on brain tissue health during lifespan, in the case of early B-vitamin deficiency. In addition, evaluation of the cleaved caspase-3 expression showed that the age-related physiological cell death appeared not modified neither by the initial deficiency nor by hypoxia. Thus, the aging process seemed to regularly occur in all animal groups, so the putative protective aspect of the hypoxia-stimulated-mechanism appeared not related to an inhibition of senescence. We obtained positive indications exploring functional plasticity in terms of level of expression of typical hippocampus synaptic actors. Globally, the level of expression of the N-methyl D-aspartate receptor 1 (NMDAR1) and the post-synaptic density-95 (PSD-95) were negatively correlated to the deficient diet alone, and positively correlated to hypoxia. This indicates that the deficiency in B-vitamins affected the hippocampus status in terms of circuit plasticity and that this aspect was maintained during the lifespan, as the Barker theory described for developmental or neonatal events leading to long term positive or negative effects [45,46]. At the same time, the hypoxia-stimulated-mechanism occurring in the same neonatal period could give advantages to receptive circuits such as the hippocampus leading to maintain a long term efficient plasticity even under transient deficiency in B-vitamins. We of course do not argue in favor of a treatment based on hypoxia in deficient infants, but our results suggest the exploration of putative routes to enhance neurogenesis and/or plasticity already shown to be latent in the brain. Thus, our conclusion is that environmental or hormonal or pharmacological stimulations of brain plasticity could act as complements with vitamin treatments for brain rescue [47,48,49,50]. In that sense, neurotrophic factors seem to represent key physiological actors at the interface between stimulation and neuronal survival [51,52,53].

## 4. Methods

### 4.1. Animal Treatments

Animal experiments were performed on Wistar rats (Charles River, l’Arbresle, France) and were conducted in accordance with the National Institutes of Health Guide for the Care and Use of Laboratory Animals, in an accredited establishment (Inserm U1256), according to governmental guidelines N86/609/CEE, and the authorization number 5,454,722; 2017 march the 31th (Local Committee for Ethics). Adult female rats were maintained under standard laboratory conditions, on a 12-hour light/dark cycle, with food and water available ad libitum. One month before pregnancy, they were fed with either standard food (*n* = 8) (Maintenance diet M20; Scientific Animal Food and Engineering, Villemoisson-sur-Orge, France) or with a diet lacking methyl donors, i.e., vitamin B12, and folate (methyl donors deficiency (MDD), *n* = 8) (Special Diet Service, Saint-Gratien, France), described previously [16]. The assigned diet was constantly maintained until weaning (i.e., postnatal day 21/PN21). At weaning, all animals received a non-deficient diet (normal diet) during the rest of their lifespan (330-days-of-age). Vitamin B6 was provided in the diet at normal levels in both groups. Methionine levels were 0.43% in both diets and homocysteine was not detectable. All experiments were conducted on pups from mothers fed either a normal or a deficient diet. Litters were reduced to 10 pups for homogeneity.

### 4.2. Exposure to Hypoxia

As published already, a mild transient hypoxia often occurs naturally during birth, and as a physiological mechanism, neonates recover their ventilation after a couple of minutes usually without damage. Thus, we used in this study, as a simple method, a mild transient hypoxia to stimulate the latent neurogenesis in the brain [23,24,25]. Within 24 h after birth, some pups were randomly chosen within a litter and placed for 5 min in a thermoregulated Plexiglas chamber flushed with 100% N_2_, whereas the remaining pups were taken as controls and exposed for the same time to 21% O_2_/79% N_2_ (a mixture corresponding to normal air). The temperature inside the chamber was adjusted to 36 °C to maintain body temperature in the physiological range. All pups were allowed to recover for 20 min in normoxic conditions and were then returned to their dams.

### 4.3. Biochemical Analyses

Plasma concentrations of vitamin B12 and folate were determined by a radio-dilution isotope assay (simulTRAC-SNB, ICN, Costa Mesa, CA, USA). Homocysteine, Methylmalonic acid and Succinic acid concentrations were measured by High Performance Liquid Chromatography (Waters, St. Quentin, France) coupled to mass spectrometry (API 4000 Qtrap Applied Biosystems, Courtaboeuf, France). The measurements of S-Adenosine Methionine (SAM) and S-Adenosine Homocysteine (SAH) concentrations were performed as described [16]. Proteins were precipitated with 0.2 M HCLO_4_, centrifuged, and the supernatant filtered through 0.45 μm before injection onto the column (LiChrospher, 100 RP-C18, 5 μm, 250 × 4 mm I.D.).

### 4.4. Behavioral Test

Animals (*n* = 10 to 15 per group) were tested for learning and memory (reference memory) in a multiple T-maze with 6-choice points and dimensions of 180 × 110 cm. The alley through which the animals navigated had a height of 35 cm and a width of 8 cm. Rat pups before weaning were trained two times per day for five consecutive days between postnatal days 14 and 18, using the motivation to reach the maternal box as the performance test. In order to test adult rats after weaning, it was not possible to use the maternal box as a motivation. However, it is assumed that transient food deprivation motivates animals to reach the goal box where they would be rewarded with food. Adult rats deprived of food for 24 h were tested for memory retrieval in the same maze at various ages, from 80 to 330 days, by two consecutive trials at each chosen time point. Time to reach the goal as well as wrong decisions at the choice points (number of errors) were recorded. For homogeneity, tests were always performed between 9:00 and 12:00 a.m.

### 4.5. Histopathological Measurements

Hippocampus tissue investigation was conducted with 4 animals per group at 330-days-of-age. Brains were quickly harvested, frozen in methylbutane at −30 °C, and kept at −80 °C. Twelve-µm sagittal cryo-sections were generated, starting from the zero plane that bisects the brain mid-sagittally, and brain structures were identified according to the Paxinos and Watson atlas for slide standardization. For subsequent staining and labeling counts, selected slides were coded prior to analysis and the codes were not broken until the experiments were completed. Ammon’s Horn hippocampus (CA1) layer thickness was measured stereologically after thionin or DAPI staining. Images were collected under 20× magnification with an Olympus microscope connected to a digital calibrated camera.

### 4.6. Quantification of Proteins

Western blot analyses were performed on nitrogen frozen isolated brain structures. Tissue was solubilized in RadioImmunoPrecipitation Assay (RIPA) Lysis buffer containing 140 mM NaCl, 0.5% (*w/v*) sodium deoxycholate, 1% (*v/v*) Nonidet P-40, 0.1% (*w/v*) SDS, and protease inhibitors (Complete, Roche Applied Science, Meylan, France). After homogenization, samples were lysed by three cycles of freezing/thawing and finally centrifuged at 4 °C for 30 min at 15,000× *g*. The protein concentration in the supernatant was determined using the BCA protein assay kit (Pierce, Interchim, Montluçon, France). Moreover, 40 µg protein samples were mixed with an equal volume of 2× Laemmli buffer, denatured by heating the mixture for 5 min at 100 °C, and then resolved by 12% SDS-PAGE. The separated proteins were transferred using a Mini Trans-Blot cell onto polyvinylidene fluoride membrane (Immobilon-P, Millipore), and the membranes were blocked for 1 h with Tris-buffered saline (pH 7.4) and 0.1% (*v/v*) Tween 20 (Tris-Buffered Saline Tween TBST buffer) containing 5% (*w/v*) bovine serum albumin. The polyvinylidene fluoride membranes were then incubated overnight at 4 °C with a primary antibody against one of the following proteins: N-methyl D-aspartate receptor 1 (NMDAR1) (Pa5-34599, rabbit polyclonal, 1/1000, Invitrogen Thermo Fisher Scientific, Waltham, MA, USA), post-synaptic density-95 (PSD-95) (ab13552, mouse monoclonal, 1/700, Abcam), cleaved caspase-3 (asp175, rabbit monoclonal, 1/700, Cell Signaling Technology, Danvers, MA, USA), glyceraldehyde-3-phosphate dehydrogenase (GAPDH, mouse monoclonal, 1/1,000, Abcam, Cambridge, UK) was used as an internal standard. Polyvinylidene difluoride membranes were incubated for 1 h at room temperature with the corresponding horseradish peroxidase-conjugated pre-adsorbed secondary antibody (1/5000, Molecular Probes, Eugene, OR, USA). Quantity One software, associated with the VersaDoc imaging system (Model 1000, Bio-Rad Laboratories, Hercules, CA, USA), was used to quantify the signals.

## Figures and Tables

**Figure 1 ijms-21-08008-f001:**
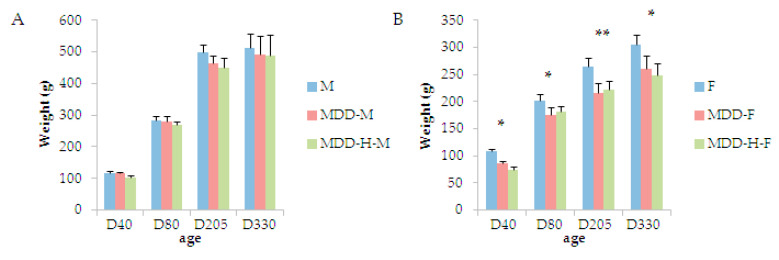
Body weights in grams of growing (**A**) male (M) and (**B**) female (F) rats after weaning (21 days of age) and the switch from the deficient (MDD for methyl donors deficiency) to the normal diet (H represent neonatal mild transient Hypoxia, *n* = 10 to 15 animals per group). Statistics, * *p* < 0.05, ** *p* < 0.01.

**Figure 2 ijms-21-08008-f002:**
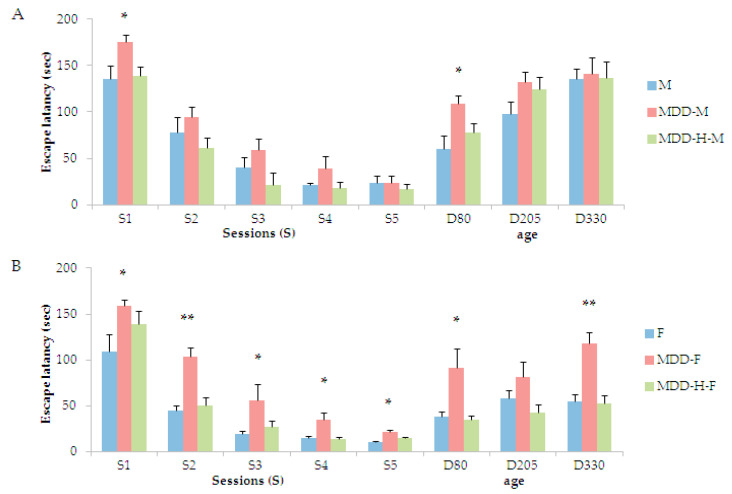
Escape latencies in seconds of (**A**) males (M) and (**B**) females (F) during the learning procedure between D14 and D18 noted as “sessions” (S1 to S5), and at three time points for memory retrieval during adult life (D80, D205, and D330, *n* = 10 to 15 animals per group). Statistics, * *p* < 0.05, ** *p* < 0.01.

**Figure 3 ijms-21-08008-f003:**
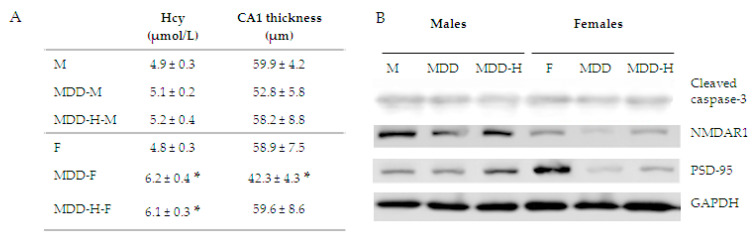
(**A**) Plasma homocysteine in µmol/L (*n* = 10 to 15 animals per group) and measurement of hippocampus CA1 layer thickness in µm at 330-days-of-age in each animal groups (*n* = 5 animals per group; Statistics, * *p* < 0.05). (**B**) Western blot analysis of protein expressions in hippocampus extracts at 330-days-of-age in each animal groups (*n* = 3 to 5 animals per group, run in duplicate). Cleaved caspase-3, N-methyl D-aspartate receptor-1 and post-synaptic density-95 were analyzed in 4 animals per group, run in duplicate, and compared with GAPDH as internal standard. The blot shown in the figure presents the representative variations observed in all blots. See the decreased expression of NMDAR1 and post-synaptic density-95 (PSD-95) in MDD groups, the restoration of their expression with hypoxia (H) despite MDD, while the level of expression of the cleaved caspase-3 remained comparable. Finally, we explored some molecular markers to evaluate tissue health of the hippocampus area at D330. We found that the physiological mortality represented by the expression of the cleaved-caspase-3 appeared similar in all groups. However, western blot analysis showed that expression of NMDAR1 and PSD-95 appeared lower in MDD males and females compared to their controls respectively. These markers were over-expressed in hypoxia groups (MDD-H males and females) compared to MDD groups in respect to gender.

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
