# Peer review of "The Stimulation of Neurogenesis Improves the Cognitive Status of Aging Rats Subjected to Gestational and Perinatal Deficiency of B9–12 Vitamins"

_ijms, 2020, doi:10.3390/ijms21218008_

Round 1

Reviewer 1 Report

The manuscript as written and presented, is difficult to follow. Needs rewriting with special attention to the written English and sequence of findings. The discussion also needs to follow a sequence that is easy for readers to follow.

The figure with the bar graphs, needs labels in the axis. The number of animals should be indicated in the figure or legend.

Author Response

The manuscript as written and presented, is difficult to follow. Needs rewriting with special attention to the written English and sequence of findings. The discussion also needs to follow a sequence that is easy for readers to follow.

We tried to improve the sequence and global writing. We chose a "short format" in order to avoid a putative dilution of the scientific message with numerous additional data. In the same aspect, we chose a condensed introduction to drive readers to the objective. If more details are still needed, we can switch the introduction to the longer format. Nevertheless, the scientific knowledges in the fields of B-vitamins deficiency and brain "natural" neurogenesis are well documented. Thus, our manuscript uses over fifty references for a "short communication" format.

The discussion section presents now sub-sections in order to improve the writing sequence and the global comprehension.

Also, we asked our colleague Pr. Simon Thornton, native English, to check the grammar and style of the manuscript; many thanks to him. We hope that it is now satisfying.

The figure with the bar graphs, needs labels in the axis. The number of animals should be indicated in the figure or legend.

Corrections are proposed in the revised version.

Reviewer 2 Report

I have some serious concerns about this study.

1)

the construction of the manuscript is very strange. The methods are located at the end and there are no conclusions. Normally the methods are described before the results, and after a discussion the conclusions are coming.

2) I don't understand why a deficiency of vitamins B9-B12 should be treated by the brain stimulation with hypoxia. The authors don't even mention the possibility to treat the animals with the adequate doses of vitamins B. And it would be the easiest way to improve the brain status if the experimental rats.

3) Because of the above I have a doubt related to the ethical aspects of this study. The rat pups were probably suffering in the experiment without any strong reason for it. And there is no statement about the ethical agreement for this study. It is a big mistake, in my opinion.

4) the study is written in a very unfriendly manner. The language is jargon, difficult, understandable only for strict specialists. There is a lack of conclusions for the practice, especially the possibility of transposing the results into human physiology.

Please try to improve the above shortages. Otherwise the manuscript cannot be published, I think.

Author Response

I have some serious concerns about this study.

1) the construction of the manuscript is very strange. The methods are located at the end and there are no conclusions. Normally the methods are described before the results, and after a discussion the conclusions are coming.

We first proposed a "short communication" in order to focus on the scientific message represented by "a complementation of the classical treatment with vitamin injections, using a stimulation of neurogenesis by various usable methods (environmental or hormonal or pharmacological stimulation of neurogenesis)". In this case, "the methods" are rather usually presented in "additional or complementary material". But it seems that such a proposed format led to confusions, so we return to a classical format. Hope to clarify the situation.

2) I don't understand why a deficiency of vitamins B9-B12 should be treated by the brain stimulation with hypoxia. The authors don't even mention the possibility to treat the animals with the adequate doses of vitamins B. And it would be the easiest way to improve the brain status if the experimental rats.

3) Because of the above I have a doubt related to the ethical aspects of this study. The rat pups were probably suffering in the experiment without any strong reason for it. And there is no statement about the ethical agreement for this study. It is a big mistake, in my opinion.

Response for remarks 2) and 3)

The study was positively evaluated by the committee for ethics. It is mentioned in the revised version.

We often receive such comments because the logical treatment for a nutritional or a genetic deficiency is represented by injection of vitamins. This is of course what patients have, as we highlighted in (i) the abstract (see the two sentences below) and (ii) the introduction and discussion sections. But for neurological outcomes, such injections are not efficient.

" ...despite the usual treatment based on B-vitamin injections, the neurological outcomes remain poorly rescued in the majority of cases",

and "Our conclusion was not focused on the neonatal hypoxia as a putative treatment, but we discussed that in case of neurologic retardation associated with a reduced B-vitamins status, the stimulation of the latent neurogenesis in infants could ameliorate the quality of life during lifespan."

Secondly, the use of an hypoxic condition is not to promote a putative treatment, of course. We have already published, like others, on that topic since numerous years (Grojean et al. Hippocampus 2003; Daval et al. Pediatric Res. 2004;Daval&Vert Semin. Perinatol. 2004; Blaise et al. Pediatric Res. 2005; Pourié et al. Neuroscience 2006; Blaise et al. Am. J. Physiol. Endocrinol. Metab. 2007; Blaise et al. Exp. Neurol. 2009; Martin et al. Semin. Perinatol. 2010; Martin et al. PLoS One, 2012). In our different publications, we insisted that a mild transient hypoxia often naturally occurs during birth, and as a physiological mechanism, neonates recover their ventilation after a couple of seconds or minutes usually without damages. Thus, we explored the neurological outcomes of hypoxic events within the first 24h of life, finding a plastic reaction of the brain. So, taking in account all these aspect, we presently used, as a simple method, a mild transient hypoxia to stimulate the latent neurogenesis in the brain; not more. This is clearly exposed in the beginning of the "method" section ("Exposure to hypoxia": highlighted in yellow).

4) the study is written in a very unfriendly manner. The language is jargon, difficult, understandable only for strict specialists. There is a lack of conclusions for the practice, especially the possibility of transposing the results into human physiology.

We are surprised with the remark concerning the lack of conclusion. The last sentences of both the Abstract and Discussion sections seemed sufficiently indicative for scientists in the field to explore putative routes for brain plasticity and neurogenesis.

" This indicates that the deficiency in B-vitamins affected the hippocampus status in terms of circuit plasticity and that this aspect were maintained during lifespan, as Barker-theory described for developmental or neonatal events leading to long term positive or negative effects [45,46]. In the same time, the hypoxia occurring in the same neonatal period could give advantages to sensitive circuit such as the hippocampus in terms of putative neurogenesis leading to maintain a long term efficient plasticity even under transient deficiency in B-vitamins. We of course do not argue in the favor of a treatment based on hypoxia in deficient infants, but our results suggest to explore putative routes to enhance the neurogenesis and/or plasticity already shown to be latent in the brain. Thus, environmental or hormonal or pharmacological stimulations could act as complements with vitamin treatments for brain rescue [47-50]."

Nevertheless, we add putative pathways/actors that could lead to such beneficial aspects.

"... stimulations could act as complements with vitamin treatments for brain rescue [47-50], and neurotrophic factors seem to represent key physiological actors at the interface between stimulations and neuronal survival [51-53].

Please try to improve the above shortages. Otherwise the manuscript cannot be published, I think.

Thank you for your review. Authors would like you to focus on the main concern of the present communication. In case of B-vitamin deficiency, the "natural" and classical treatment is always based on injections of vitamins in order to correct the physiological status. But the brain tissue and functioning still keep deleterious outcomes during lifespan, especially for learning, memory and other cognitive functions (hippocampus-related). We would highlight that it could be interesting to add to the classical treatment, a stimulation of brain neurogenesis or plasticity. Various methods could be tested, such as pharmacological up-regulation of neurotrophic factors, or hormonal activation of cell receptors for example (this is added at the end of the discussion).

Importantly, we asked our colleague Pr. Simon Thornton, native English, to check the grammar and style of the manuscript; many thanks to him. We hope that it is now satisfying.

We tried to improve each part of our manuscript.